# Prediction of the Adult T-Cell Leukemia Inhibitory Activity of Blueberry Leaves/Stems Using Direct-Injection Electron Ionization-Mass Spectrometry Metabolomics

**DOI:** 10.3390/plants11101343

**Published:** 2022-05-19

**Authors:** Hisahiro Kai, Yoshihito Okada, Yo Goto, Takayuki Nakayama, Kazuhiro Sugamoto, Kenjirou Ogawa, Masao Yamasaki, Kazuhiro Morishita, Koji Matsuno, Hisato Kunitake

**Affiliations:** 1Department of Pharmaceutical Health Sciences, School of Pharmaceutical Sciences, Kyushu University of Health and Welfare, 1714-1 Yoshino-Machi, Nobeoka, Miyazaki 882-8508, Japan; 2Department of Natural Medicine and Phytochemistry, Meiji Pharmaceutical University, 2-522-1 Noshio, Kiyose, Tokyo 204-8588, Japan; y-okada@my-pharm.ac.jp; 3Biolabo Co., Ltd., 7-2-6 Minatojimaminamimachi Chuo-ku, Kobe, Hyogo 650-0047, Japan; ygoto@ge-hd.co.jp (Y.G.); tnakayama@ge-hd.co.jp (T.N.); 4Department of Applied Chemistry, Faculty of Engineering, University of Miyazaki, 1-1 Gakuen Kibanadai-nishi, Miyazaki 889-2192, Japan; sugamoto@cc.miyazaki-u.ac.jp; 5Organization for Promotion of Tenure Track, University of Miyazaki, Miyazaki 889-2192, Japan; ogawa.kenjirou.u2@cc.miyazaki-u.ac.jp; 6Department of Biochemistry and Applied Biosciences, Faculty of Agriculture, University of Miyazaki, 1-1 Gakuen Kibanadai-nishi, Miyazaki 889-2192, Japan; myamasaki@cc.miyazaki-u.ac.jp (M.Y.); hkuni@cc.miyazaki-u.ac.jp (H.K.); 7Department of Medical Sciences, Division of Tumor and Cellular Biochemistry, Faculty of Medicine, University of Miyazaki, 5200 Kihara, Kiyotake, Miyazaki 889-1692, Japan; kmorishi@med.miyazaki-u.ac.jp; 8Department of Hygiene, School of Pharmaceutical Sciences, Kyushu University of Health and Welfare, 1714-1 Yoshino-machi, Nobeoka, Miyazaki 882-8508, Japan; kjmtsn@phoenix.ac.jp

**Keywords:** metabolomics, direct-injection electron ionization-mass spectrometry (DI-EI-MS), *Vaccinium virgatum*, adult T-cell leukemia (ATL)

## Abstract

Although *Vaccinium virgatum* Aiton leaves and stems inhibit adult T-cell leukemia (ATL) cells, leaves and stems can differ between individual plants and by time and location. In this study, leaf and stem components were profiled in the same individual plant using direct-injection electron ionization-mass spectrometry (DI-EI-MS) metabolomics, with the aims of analyzing the anti-ATL activity, and quantifying proanthocyanidins (PACs). Leaves, stems, and leaf/stem mixtures showed distinct and characteristic spectra. Anti-ATL activity was stronger in stems than leaves, and the PAC content was higher in stems than leaves. These data were subjected to bivariate analysis to identify the factor (*m/z*) responsible for the inhibitory effect of ATL based on the highest coefficient of determination (*R*^2^). The results of this DI-EI-MS metabolomics analysis suggest that among PACs contained in *V. virgatum* stems and leaves, the fragment ion at *m/z* 149 contributes significantly to anti-ATL activity.

## 1. Introduction

Multiple species are involved in the commercial production of blueberries, with the vast majority composed of *Vaccinium corymbosum* L. (highbush blueberry) and its hybrids and *V. angustifolium* Aiton (lowbush blueberry), with lesser quantities of *V. virgatum* Aiton (rabbit-eye blueberry) [1,2]. *V. virgatum* is known to be suitable for cultivation in warm regions [3]. Blueberry fruit are widely used in traditional medicine, as well as the pharmaceutical and food industries [4,5,6]. The fruit of *V. virgatum* are edible and are used in sauces and syrups, as well as bread, muffins, pancakes, and pies. Recent studies have revealed that components of *V. virgatum* leaves suppress lipid accumulation and uric acid production in adipocytes [7], affect alcohol metabolism during chronic ethanol consumption [8], exhibit anti-fibrogenic properties [9], and suppress replication of the hepatitis C virus [10]. A 90-day subchronic toxicity study in rats revealed that *V. virgatum* leaves are safe for consumption [11]. In addition, we reported that *V. virgatum* leaves prevent the proliferation of adult T-cell leukemia (ATL) cells [12,13]. ATL is an aggressive chemotherapy-resistant malignancy secondary to infection with the retrovirus human T-cell leukemia virus type 1 (HTLV-1) [14]. Based on these studies, agricultural efforts have focused on breeding *V. virgatum* leaves for use as green tea, although the leaves have not previously been used as a food ingredient. Tetsumura et al. established a new species of *V. virgatum* known as “Kunisato 35 Gou” for leaf cultivation [15], and Yamasaki et al. reported that the leaves contain large amounts of proanthocyanidin (PAC) [7]. With regard to the chemical constituents of blueberry leaves, Wu et al. reported the total phenolic, flavonoid, and PAC contents of leaf extracts from 73 blueberry cultivars [16]. Matsuo et al. elucidated the structure of a novel PAC isolated from a methanol extract of blueberry leaves [17]. These reports suggest that PACs play a role in the biological activities of blueberry leaves. In the first reported instance of identified PAC from a *V. virgatum* leaf [10], the methanol extract was fractionated using organic solvent extraction and high-performance liquid chromatography (HPLC). The structure of PAC was then identified by liquid chromatography/mass-ion trap-time of flight analysis and butanol-HCl hydrolysis analysis. On the other hand, it was confirmed that PAC was present on the *V. virgatum* stem by the 4-dimethylaminocinnamaldehyde (DMACA) method in our previous report [18].

Interestingly, a study using a direct-injection electron ionization-mass spectrometry (DI-EI-MS) metabolomics method established a relationship between blueberry species and ATL-related bioactivity and seasonal variation [19]. DI-EI-MS is a multivariate analytical method useful for characterizing biological materials [20]. Metabolite fingerprinting approaches based on MS can be used to characterize metabolite profiles of complex plant extracts. Wu et al. used a metabolomics approach to investigate the metabolites present in the leaves and stem bark of dioecious *Morus alba* L., analyzing the differential metabolites of male and female plants by combining the approach with chemometrics, and also evaluating the antioxidant activity of *Morus alba* L. [21].

More recently, we conducted a quantitative analysis of water extracts of *V. virgatum* stems and found that these plants are rich in polyphenols (included PACs), exhibiting antioxidant activity and inhibiting ATL cell growth [18]. However, there are no reports of studies comparing the PAC content and functionality of leaves and/or stems in the same individual plant to evaluate the potential for the industrial utilization of blueberry stems and leaves. The value of blueberry stems can be determined using metabolomics methods, which are excellent tools for comprehensive analyses. It is important not only that leaves and stems be evaluated separately, but also that mixtures of leaves and stems (i.e., aerial parts) of blueberry plants be evaluated in terms of their potential use as more rational functional materials. DI-EI-MS metabolomics methods are ideal tools for characterizing plant organs within the same individual. In the present study, we evaluated the *V. virgatum* leaf/stem functionality for the prevention of ATL by DI-EI-MS metabolomics.

## 2. Results

### 2.1. DI-EI-MS Profiling of Blueberry Leaves and Stems

Water extracts of *V. virgatum* leaves, stems, and leaf/stem mixtures were prepared from 10 individual plants (Table 1). Figure 1 shows a typical DI-EI-MS spectra of leaves, stems, and leaf/stem mixtures. The mass spectra of the 10 samples were almost the same, indicating that while there are differences between organs, the differences between individuals are small. The top three peaks were observed as follows: *m/z* 110, 61, and 73 for leaves (Figure 1A); *m/z* 61, 73, and 58 for stems (Figure 1B); and *m/z* 61, 73, and 110 for leaf/stem mixtures (Figure 1C). Many other characteristic fingerprint fragments were observed in each sample type.

### 2.2. Inhibition of ATL-Related Cell Growth

In our most recent study, we analyzed the fractions obtained from water extracts of *V. virgatum* stems for the ability to inhibit the proliferation of ATL cell lines [18]. However, questions regarding which organ contains the most active components, or which is stronger when mixed, have not been rigorously tested in individual plants. Figure 2 shows the effect of each extract or epigallocatechin gallate (EGCG) at 50 μg/mL on inhibiting the proliferation of three different ATL cell lines (ED, S1T, and Su9T01). EGCG was used as a positive control because it exhibits inhibitory effects against ATL cells in the same assay [22]. The activity values (% of control) of EGCG against ED, S1T, and Su9T01 cells were 24 ± 5%, 18 ± 6%, and 14 ± 1%, respectively. All samples significantly inhibited the proliferation of all three ATL cell lines. The activity values for leaves, stems, and leaf/stem mixtures were as follows: 24 ± 14%, 12 ± 4%, and 27 ± 6% against ED cells (Figure 2A); 23 ± 9%, 11 ± 8%, and 35 ± 4% against S1T cells (Figure 2B); and 5 ± 5%, 2 ± 1%, and 7 ± 2% against Su9T01 cells (Figure 2C), respectively. These results indicate that stems are more active than leaves in inhibiting the proliferation of the three ATL cell lines. In contrast, mixed leaves and stems showed only weak activity compared with leaves or stems alone.

### 2.3. Prediction of ATL Inhibitory Activity Using DI-EI-MS Metabolomics

A bivariate analysis of DI-EI-MS data was performed for the purpose of predicting ATL inhibitory activity based on the intensity of characteristic *m/z* peaks. As the measurement range in the DI-EI-MS method was set to 50–500 *m/z*, 451 bivariate analysis results were obtained per ATL cell line (ED, S1T, and Su9T01). Linear regression analysis was performed on the 451 analytes, which were sorted in descending order of *m/z* with the highest coefficient of determination (*R*^2^). Interestingly, a peak at approximately *m/z* 149 exhibited the highest *R*^2^ value common to all three ATL cell lines, and this was reproducible in three repeated individual experiments (Table 2). These results suggest that the molecular weight (or formula weight) of the metabolite or its partial structure that contributes to this biological activity is likely to be 149 or 148. In addition, the three repeated experiments revealed another peak at *m/z* 131 that may also be involved in inhibitory activity (Table 2). Figure 3 shows the results of the bivariate analysis with ATL cell proliferation inhibitory activity as the objective variable and peak intensity at particular *m/z* values as the explanatory variable.

### 2.4. Analysis of PACs

PACs are polyphenols that exhibit a variety of biological activities, and previous reports suggest that *V. virgatum* leaves and stems are rich in PACs [17,18]. However, these previous studies differed in terms sampling area and time. Hence, in the present study, we analyzed PACs in *V. virgatum* leaves, stems, and leaf/stem mixtures. Figure 4 shows the PAC content of water extracts of *V. virgatum* leaves, stems, and leaf/stem mixtures from the same individual plants (using 10 individual plants, *n* = 10), analyzed using the 4-dimethylaminocinnamaldehyde (DMAC) method. Stems contained the highest PAC content, at 77.7 ± 10.7 mg/g in terms of substantial quantity of catechin per dry weight of extract. This amount was approximately three times the amount of PACs found in leaves (27.7 ± 6.4 mg/g). The amount of PACs found in the leaf/stem mixtures was 40.4 ± 7.4 mg/g, intermediate between leaves and stems.

### 2.5. Three Correlation Matrices for ATL Cell Inhibitory Activity and DI-EI-MS and PAC Analyses

A correlation matrix was created to confirm whether the PAC content was associated with the peak intensity at *m/z* 149 and the ATL cell inhibitory activity. Figure 5 shows scatterplot matrices for inhibitory effect against ATL cells, the peak intensity at *m/z* 149 on DI-EI-MS analysis, and the PAC content. The correlation coefficient (*r*) value between the PAC content and the peak intensity at *m/z* 149 was 0.8107, indicating a relatively strong positive correlation. The *r* values between the PAC content and inhibitory activity against ED, S1T, and Su9T01 cells were 0.6112, 0.6202, and 0.5278, respectively (Figure 5A–C). These three *r* values were similar. However, the *r* values between peak intensity at *m/z* 149 and inhibitory activity against ED, S1T, and Su9T01 cells were 0.5933, 0.7437, and 0.5141, respectively (Figure 5A–C). The magnitude of the relationship between these *r* values was the same as that for the coefficient of determination (*R*^2^) values, which were 0.3520, 0.5531, and 0.2643 for ED, S1T, and Su9T01 cells, respectively (Figure 3A–C). Although the degree of correlation differed depending on the cell line, these data indicate that ATL cell inhibitory activity can be predicted not only by the amount of PAC but also by DI-EI-MS metabolomics analysis.

## 3. Discussion

In this study, *V. virgatum* leaves, stems, and leaf/stem mixtures were profiled using a DI-EI-MS method, and the results were analyzed in conjunction with assays of inhibition of ATL cell proliferation to develop a prediction model based on DI-EI-MS spectra. PACs are oligomers or polymers of flavan-3-ol molecules linked via inter-flavan linkages. Oligomeric and polymeric PACs predominantly consist of catechin and epicatechin monomers [23,24]. As PACs are polymers, they cannot always be represented by a constant molecular weight or molecular formula. Moreover, it is very difficult to isolate and purify the compounds from natural sources containing large amounts of PACs, such as blueberry leaves and stems. However, metabolomics is a powerful tool for evaluating the quality of natural sources containing functional components that are difficult to isolate and identify, such as PACs. We expected the leaves and stems of *V. virgatum* to contain the same amount of PACs, but the spectra obtained from this metabolomics analysis indicated clear characteristic differences between the organs. Furthermore, it was possible to derive an *m/z* value associated with ATL cell inhibitory activity. These results suggest that the molecular weight (or formula weight) of the metabolite (or its partial structure) that contributes to this biological activity is likely to be 149. There are no reports about DI-EI-MS metabolomics for other plant materials and their composition and biological activity. However, some analyses of phenolic compounds and pharmacological activity by LC-MS metabolomics have been reported [25,26]. This is the first study using metabolomics for functional comparison and molecular identification in the same individual *V. virgatum* plants. We developed a new method for this purpose using a DI-EI-MS metabolomics approach. In the future, we would like to make similar comparisons of the functionality of *V. virgatum* with regard to activities other than the inhibition of ATL cell proliferation [7,8,9,10], and verify the predictive power of DI-EI-MS metabolomics with regard to assessments of functionality.

## 4. Materials and Methods

### 4.1. Plant Material

Leaves and stems of *V. virgatum* (Kunisato 35 Gou) were collected in November and December 2020 in Yaeokobo (Kunitomi, Miyazaki, Japan; 32°02′36″ N., 131°16′50″ E.). The number of voucher specimens is indicated in Table 1, and the specimens were deposited in the Department of Pharmaceutical Health Sciences, School of Pharmaceutical Sciences, Kyushu University of Health and Welfare (Nobeoka, Miyazaki, Japan). Ten blueberry plants that had been sufficiently air-dried were divided into leaves and stems, and the yield of each was measured. For the mixed samples, the leaves and stems were combined based on the ratio of the measured yields. For example, in the case of an individual having a 1.0 g dry weight of leaves and a 2.0 g dry weight of stems, a sample in which leaves and stems were mixed at a ratio of 1:2 was prepared.

### 4.2. Extraction

The extraction method was a slight modification of our previous report [13]. Raw samples of *V. virgatum* leaves, stems, and leaf/stem mixtures (approximately 10 g) were dried thoroughly at room temperature and extracted using H_2_O (300 mL for 1 h, once) at 60 °C. The water extracts were naturally filtered, and the filtrates were freeze-dried. The yields of the freeze-dried powder from the water extracts are shown in Table 1.

### 4.3. DI-EI-MS

DI-EI-MS analysis was performed according to our previous report [20]. In the present study, the electrons were accelerated at 70 eV in the region between the filament and the entrance to the ion source block. EI-MS was conducted using a double-focusing mass spectrometer (JMS GC-mate II; JEOL, Tokyo, Japan) equipped with a heated direct-injection sample probe. The MS detector parameters used were as follows: interface temperature, 320 °C; ion-source temperature, 280 °C; ionization mode, EI; electron energy, 70 eV; scan speed, 0.3 s/scan; interdelay, 0.2 s; scan range, *m/z* 50−500; accelerating voltage, 2500 V; ionization current (emission), 0.3 mA; and variable temperature duration, 0.5−2.5 min. A 1 μL aliquot of the test sample in water (5 mg/mL) was injected into the DI-EI-MS system. A microsyringe was used to load a drop (1 μL) of each extract solution onto a glass target, which was then loaded into the direct-injection sample probe (MS-DIP25). The integration analysis covered the range of approximately scans 120−360 of the total ion chromatogram (TIC). The TIC was monitored for 2.5 min, and all of the fragment ions between 0.8 and 2.5 min were added.

### 4.4. ATL Cell Proliferation Assay

ED cells were kindly provided by Dr. M. Maeda (Kyoto University, Kyoto, Japan), and S1T and Su9T01 cells were kindly provided by Dr. N. Arima (Kagoshima University, Kagoshima, Japan). [27]. ATL cell proliferation assays were conducted according to the method of a previous report [13]. All cells were maintained in RPMI 1640 medium (Sigma−Aldrich Co., St. Louis, MO, USA) supplemented with 10% fetal bovine serum (Sigma−Aldrich Co., lot no. 13C491) containing 100 U/mL penicillin G and 100 μg/mL streptomycin (Life Technologies, Carlsbad, CA, USA). Each cell line was seeded (1 × 10^5^ cells/mL, 90 μL/well) into a 96-well plate containing RPMI 1640 medium. After incubation at 37 °C for 24 h in an atmosphere containing 5% CO_2_, the samples were added (10 μL/well) to the cells and incubated for an additional 72 h. Subsequently, the inhibition of cell proliferation was determined using a 2-(2-methoxy-4-nitrophenyl)-3-(4-nitrophenyl)-5-(2,4-disulfophenyl)-2*H*-tetrazolium monosodium salt (WST-8) assay kit (Dojindo, Kumamoto, Japan). Viable cells convert the tetrazolium salt in WST-8 to highly water-soluble formazan, which is monitored by measuring the absorbance at 450 nm using a microplate reader (Multiskan FC, Thermo Fisher Scientific, Waltham, MA, USA). EGCG (Nagara Science, Gifu, Japan) was used as a positive control [22], and was kindly provided by Dr. J. Sonoda (Kyushu University of Health and Welfare, Nobeoka, Japan).

### 4.5. PAC Analysis

The PAC content was determined according to the DMAC protocol using a modification of the method of Oki et al. [28]. Two different assay solvents were prepared. Solvent A consisted of three components, ethanol:methanol:2-propanol (90:5:5). Solvent B consisted of solvent A and water (95:5). Forty microliters of sample or catechin standard solution (dissolved in solvent B) and 200 μL of DMAC solution (0.1%, dissolved in 1.2 M HCl-solvent A) were mixed. The amount of DMAC sample (or catechin) complex in the solution was determined by measurement of the optical density at 620 nm after incubation for 15 min at 30 °C. The standard curve was linear between 1.25 and 20 μg/mL of catechin. Each dried extract was dissolved in water for adjustment to 50 mg/mL. The leaf extracts were diluted 10-fold, whereas stems and mixed extracts were diluted 100-fold, using solvent B in each case. The results are expressed as the substantial quantity of catechin content/dry weight of extract (mg/g). Each sample was measured three times.

### 4.6. Statistical Analysis

Results of ATL cell proliferation assays and PAC analyses were obtained from 10 extracts, and the data shown in the figures represent mean values ± standard deviation (*n* = 10). The statistical significance of the differences was evaluated using the unpaired two-tailed Student’s *t*-test (* *p* < 0.05). The data were analyzed using Microsoft Excel.

The resulting DI-EI-MS data sets were imported into JMP Pro14.2 software (SAS Institute Inc., Cary, NC, USA) for bivariate and multivariate (scatterplot matrix) statistical analysis. All statistical analyses were carried out using JMP Pro14.2 software to identify the features contributing to group separation.

## 5. Conclusions

The results of this DI-EI-MS metabolomics analysis suggest that among PACs contained in *V. virgatum* stems and leaves, the fragment ion at *m/z* 149 contributes significantly to anti-ATL activity.

## Figures and Tables

**Figure 1 plants-11-01343-f001:**
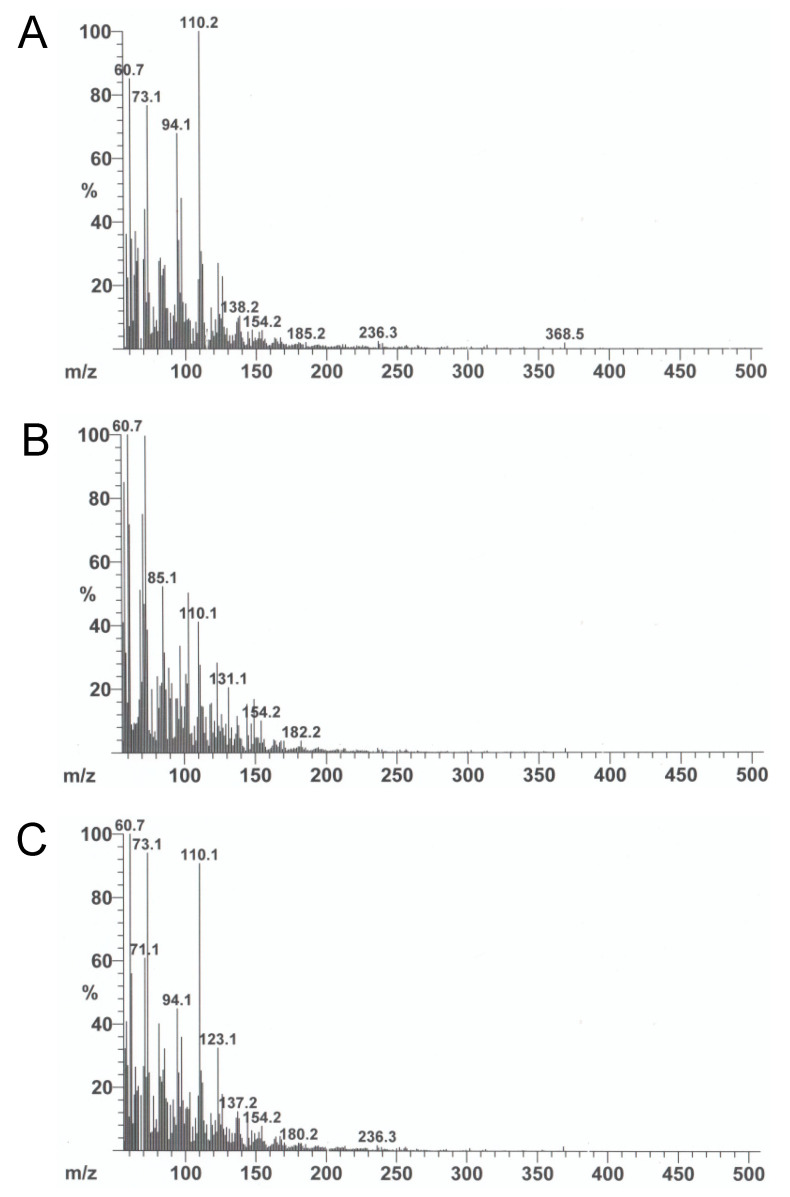
The integrated mass spectra of water extracts of *V. virgatum* aerial parts: (**A**) leaves, (**B**) stems, (**C**) leaf/stem mixtures.

**Figure 2 plants-11-01343-f002:**
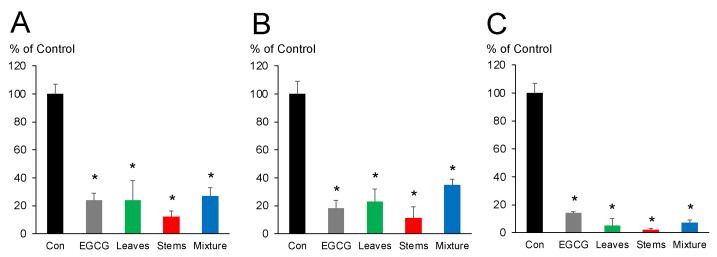
The inhibitory effects of water extracts of *V. virgatum* leaves, stems, and leaf/stem mixtures against three ATL cell lines: (**A**) ED, (**B**) S1T, and (**C**) Su9T01. The data are mean ± SD for 10 individual samples. Statistical significance was evaluated using unpaired two-tailed Student’s *t*-tests (* *p* < 0.05). Con; Control, EGCG; epigallocatechin gallate.

**Figure 3 plants-11-01343-f003:**
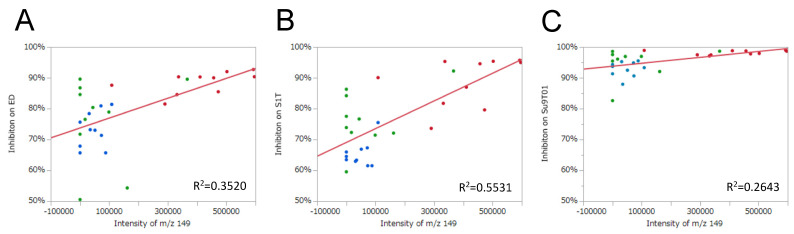
The bivariate analysis of ATL cell proliferation inhibitory activity and peak intensity at *m/z* 149 on DI-EI-MS analysis. The *m/z* 149 peak was extracted as having the highest *R*^2^ value in the measurement range *m/z* 50–500, and the results were common to all three ATL cell lines. (**A**) ED cells, (**B**) S1T cells, and (**C**) Su9T01 cells. Green circles indicate leaves, red circles indicate stems, and blue circles indicate leaf/stem mixtures.

**Figure 4 plants-11-01343-f004:**
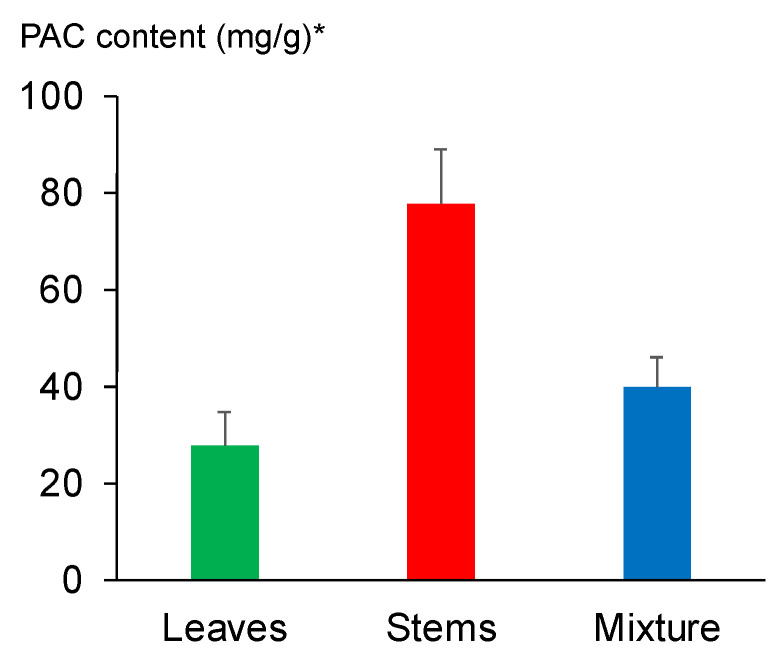
The PAC content of fractions from water extracts of *V. virgatum* leaves, stems, and leaf/stem mixtures. The data are mean ± SD for 10 individual samples. * The PAC content is substantial quantities of catechin content/dry weight of the extract (mg/g).

**Figure 5 plants-11-01343-f005:**
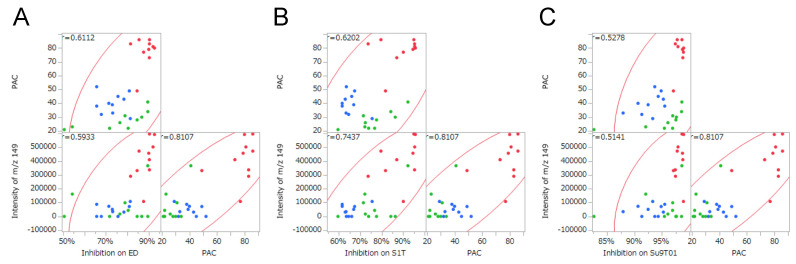
Scatterplot matrices among inhibitory effect against ATL cells, peak intensity at *m/z* 149 on DI-EI-MS analysis, and the PAC content. (**A**) ED cells, (**B**) S1T cells, and (**C**) Su9T01 cells. Green circles indicate leaves, red circles indicate stems, and blue circles indicate leaf/stem mixtures.

**Table 1 plants-11-01343-t001:** The yield of water extracts of *V. virgatum* leaves, stems, and leaf/stem mixtures.

Part	Voucher Specimen	Yield (g)
Leaves	KUHW2020BB-L1	1.30
	KUHW2020BB-L2	1.40
	KUHW2020BB-L3	1.30
	KUHW2020BB-L4	1.50
	KUHW2020BB-L5	1.70
	KUHW2020BB-L6	2.60
	KUHW2020BB-L7	2.70
	KUHW2020BB-L8	3.10
	KUHW2020BB-L9	2.50
	KUHW2020BB-L10	1.90
Stems	KUHW2020BB-S1	0.51
	KUHW2020BB-S2	0.32
	KUHW2020BB-S3	0.46
	KUHW2020BB-S4	0.40
	KUHW2020BB-S5	0.45
	KUHW2020BB-S6	0.40
	KUHW2020BB-S7	0.43
	KUHW2020BB-S8	0.41
	KUHW2020BB-S9	0.40
	KUHW2020BB-S10	0.47
Mixture	KUHW2020BB-M1	1.24
	KUHW2020BB-M2	1.43
	KUHW2020BB-M3	1.29
	KUHW2020BB-M4	1.21
	KUHW2020BB-M5	0.87
	KUHW2020BB-M6	1.06
	KUHW2020BB-M7	1.56
	KUHW2020BB-M8	1.81
	KUHW2020BB-M9	1.98
	KUHW2020BB-M10	1.84

**Table 2 plants-11-01343-t002:** The regression coefficient (*R*^2^) values for linear regression analysis of the prediction of ATL inhibitory activity using DI-EI-MS metabolomics.

ATL Cell Line	ATL Assay Trial	*R*^2^ 1st Place	*R*^2^ 2nd Place	*R*^2^ 3rd Place
ED	1	*m/z* 149	*m/z* 383	*m/z* 330
	2	*m/z* 149	*m/z* 057	*m/z* 085
	3	*m/z* 149	*m/z* 057	*m/z* 131
S1T	1	*m/z* 149	*m/z* 290	*m/z* 346
	2	*m/z* 149	*m/z* 131	*m/z* 346
	3	*m/z* 149	*m/z* 131	*m/z* 330
Su9T01	1	*m/z* 401	*m/z* 149	*m/z* 429
	2	*m/z* 235	*m/z* 298	*m/z* 226
	3	*m/z* 149	*m/z* 131	*m/z* 330

Individual experiments were repeated three times.

## Data Availability

Not applicable.

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
