# Peer review of "Prediction of the Adult T-Cell Leukemia Inhibitory Activity of Blueberry Leaves/Stems Using Direct-Injection Electron Ionization-Mass Spectrometry Metabolomics"

_plants, 2022, doi:10.3390/plants11101343_

Round 1
Reviewer 1 Report
The abstract should better represent the content and the aim of the manuscript. 2-3 lines for conclusions should be added in the abstract. Please reduce the number of keywords.
The introduction is too poor. The authors should improve this part by adding more information on the main proanthocyanidins identified in the V. virgatum leaf/stem extracts.
Did the authors use validated MS methods? It would be interesting to add an HPLC identification and quantifications of PACs in the extracts.
Results and Discussion sections should be integrated into a single paragraph to avoid redundant information and repetitions.
Conclusions should report the main issues of the manuscript. Please re-write this section.
Author Response
Answers to Reviewer: 1
Point 1: The abstract should better represent the content and the aim of the manuscript. 2-3 lines for conclusions should be added in the abstract.
Response 1: Thank you for the useful comment. The content and purpose of the manuscript was explained on line 24. Thus, we revised initial representation of line 24 from “In this study, ~~~” to “The aim of this study, ~~~”.
Point 2: Please reduce the number of keywords.
Response 2: We reduced the number of keywords from 6 to 4. We deleted “stem” and “proanthocyanidin (PAC)”.
Point 3: The introduction is too poor. The authors should improve this part by adding more information on the main proanthocyanidins identified in the V. virgatum leaf/stem extracts.
Response 3:
According to your suggestion, we added more information about identified proanthocyanidins in the V. virgatumleaf/stem extracts.
Lines 60-66: In the first reported instance of identified PAC from V. virgatum leaf [10], the methanol extract was fractionated using organic solvent extraction and high-performance liquid chromatography (HPLC). Then, the structure of PAC was identified by liquid chromatography /mass-ion trap-time of flight analysis and butanol-HCl hydrolysis analysis. On the other hand, it was confirmed that PAC was present on the V. virgatum stem by the 4-dimethylaminocinnamaldehyde (DMACA) method in our previous report [18].
Point 4: Did the authors use validated MS methods? It would be interesting to add an HPLC identification and quantifications of PACs in the extracts.
Response 4: Thank you for the interesting suggestion. This DI-EI-MS method has been validated (Reference [20]). We would like to investigate the HPLC identification and quantification of PACs in the extract in the future.
Point 5: Results and Discussion sections should be integrated into a single paragraph to avoid redundant information and repetitions.
Response 5: According to your suggestion, we revised a single paragraph. In addition, another reviewer suggestedDiscussion section to be more summarized. Since, we revised Discussion part overall.
Point 6: Conclusions should report the main issues of the manuscript. Please re-write this section.
Response 6: As described below, we revised the conclusion to be a single sentence.
Lines 286-288: The results of this DI-EI-MS metabolomics analysis suggest that among PACs contained in V. virgatumstems and leaves, the fragment ion at m/z 149 contributes significantly to anti-ATL activity.
Thank you very much for your kind suggestions.
Reviewer 2 Report
The paper aims to analyze the proanthocyanidin levels in Vaccinium virgatum stems and leaves with the use of direct-injection electron ionization-mass spectrometry metabolomics. The idea of the work seems to be very interesting. However, the experimental part lacks numerous data important for repetition of the work and verification of its scientific soundness. The work lacks also proper discussion - the obtained results are simply described with no mention of any possible implications and impact of the experimental observations. In my opinion, the submission requires extensive editions and improvement at many points, the main of which are listed below.
Major issue:
L181-194: Repeating the description of the results from the previous section of the manuscript.
L195-207: In this part of the discussion, please compare the results of the studies described in the presented manuscript with the available literature data on metabolic studies using the DI-EI-MS technique for other plant materials and their phenolic composition and biological activity.
Minor issues:
L38-39: This sentence is false. Berries of Vaccinium virgatum are edible and are used as sauces and syrups, and for bread, muffins, pancakes, and pies. Please re-edit this sentence.
L51: Please describe which groups and specific polyphenolic compounds have been identified so far in the leaves of the species under study.
L91: Please use the abbreviation “V. virgatum” in the whole manuscript.
L182: Was it really about the "weaker activity of stems compared to leaves and stems"?
L217: Please add the longitude and latitude of the crop harvest site (Yaeokobo).
L222-223: This description is incomprehensible. Please describe in detail the weight ratio in which the leaves and stems were mixed to prepare the mixed samples.
L225: “V. virgatum” written in italics. Please check the whole manuscript thoroughly.
L226: By what method were the raw samples extracted?
L247: Please describe the method of culturing the cells.
Author Response
Answers to Reviewer: 2
Major issue:
Point 1: L181-194: Repeating the description of the results from the previous section of the manuscript.
Response 1: Thank you for the suggestion. We have removed the specified part to avoid repeating the description of the result.
Point 2: L195-207: In this part of the discussion, please compare the results of the studies described in the presented manuscript with the available literature data on metabolic studies using the DI-EI-MS technique for other plant materials and their phenolic composition and biological activity.
Response 2: Thank you for the interesting suggestion. Unfortunately, there are no report about your request. However, we added the below sentence, which contains similar MS metabolomics (L 201-204).
“There are no report about DI-EI-MS metabolomics for other plant materials and their composition and biological activity. However, some examples of analysis of phenolic compounds and pharmacological activity by LC-MS metabolomics have been reported [25, 26].”
Minor issue:
Point 3: L38-39: This sentence is false. Berries of Vaccinium virgatum are edible and are used as sauces and syrups, and for bread, muffins, pancakes, and pies. Please re-edit this sentence.
Response 3: Thank you for the important suggestion. According to your suggestion, we revised the below sentence (L 43-44). “The fruit of V. virgatum are edible and are used as sauces and syrups, and for bread, muffins, pancakes, and pies.”
Point 4: L51: Please describe which groups and specific polyphenolic compounds have been identified so far in the leaves of the species under study.
Response 4: According to your comment, we revised the below sentences and added reference [7]. Here, it seems more appropriate to use proanthocyanidin (PAC) than polyphenols. L54-58: “Tetsumura et al. established a new species of V. virgatum known as “Kunisato 35 Gou” for leaf cultivation [15] and Yamasaki et al. reported that the leaves contain large amounts of proanthocyanidin (PAC) [7]. With regard to chemical constituents of blueberry leaves, Wu et al.reported the total phenolic, total flavonoid, and PAC contents of leaf extracts from 73 blueberry cultivars [16].
Point 5: L91: Please use the abbreviation “V. virgatum” in the whole manuscript.
Response 5: Thank you for the suggestion. We revised “V. virgatum” at the L 101. We checked the whole manuscript thoroughly. Overall, there are no other fixes.
Point 6: L182: Was it really about the "weaker activity of stems compared to leaves and stems"?
Response 6: According to your Point 1 and our Response 1, we deleted this sentence. We checked if there was a significant difference between stem group and mixture group. There was a significant difference on ED and S1T cells (Figure 2A and 2B), but no significant difference on Su9T01 cell (Figure 2C). Therefore, it is appropriate to delete this sentence.
Point 7: L217: Please add the longitude and latitude of the crop harvest site (Yaeokobo).
Response 7: According to your suggestion, we added the latitude and longitude information.
L214: Yaeokobo (Kunitomi, Miyazaki, Japan; 32°02‘36“ N, 131°16‘50“ E.)
Point 8: L222-223: This description is incomprehensible. Please describe in detail the weight ratio in which the leaves and stems were mixed to prepare the mixed samples.
Response 8: Thank you for the suggestion. According to your suggestion, we added the below sentence (L220-222).“For example, in the case of an individual having a dry weight of leaves of 1.0 g and a dry weight of stems of 2.0 g, a sample in which leaves and stems were mixed at a ratio of 1: 2 was prepared.”
Point 9: L225: “V. virgatum” written in italics. Please check the whole manuscript thoroughly.
Response 9: We checked the whole manuscript thoroughly. We italicized L225 and References No. 2, 6, 7, 9, 11, 13, 17, 18, 22, 25 and 26 (included other words that need to be italic).
Point 10: L226: By what method were the raw samples extracted?
Response 10: Thank you for the question. Raw samples were thoroughly dried at room temperature prior to extraction. According to your suggestion, we revised the below sentence (L224-227) “Raw samples of V. virgatum leaves, stems, and leaf/stem mixtures (approximately 10 g) were dried thoroughly at room temperature and extracted using H2O (300 mL for 1 h, 1 time) at 60°C.”.
Point 11: L247: Please describe the method of culturing the cells.
Response 11: According to your suggestion, we added the below sentence (L 248-260).
All cells were maintained in RPMI 1640 medium (Sigma-Aldrich Co., St. Louis, MO, USA) supplemented with 10% fetal bovine serum (Sigma-Aldrich Co., lot no. 13C491) con-taining 100 U/mL penicillin G and 100 μg/mL streptomycin (Life Technologies, Carlsbad, CA, USA). Each cell line was seeded (1 × 105 cells/mL, 90 μL/well) into a 96-well plate containing RPMI 1640 medium. After incubation at 37°C for 24 h in an atmosphere containing 5% CO2, samples were added (10 μL/well) to the cells and incubated for an additional 72 h. Subsequently, the inhibition of cell proliferation was determined using a 2-(2-methoxy-4-nitrophenyl)-3-(4-nitrophenyl)-5-(2,4-disulfophenyl)-2H-tetrazolium monosodium salt (WST-8) assay kit (Dojindo, Kumamoto, Japan). Viable cells convert the tetrazolium salt in WST-8 to highly water-soluble formazan, which is monitored by measuring the absorbance at 450 nm using a microplate reader (Multiskan FC, Thermo Fisher Scientific, Waltham, MA, USA).
Thank you very much for your kind suggestions.
Reviewer 3 Report
The present manuscript investigates the composition of the leaf and stem components of Vaccinium virgatum Aiton using direct-injection electron ionization-mass spectrometry (DI-EI-MS) metabolomics, followed by the analysis of anti-adult T-cell leukemia activity, and quantification of proanthocyanidins. The data were subjected to bivariate analysis to identify the factor responsible for the inhibitory effect of adult T-cell leukemia based on the highest coefficient of determination.
The main outcome of the study is related to the contribution of the (E)-3-(3,4-dihydroxyphenyl)allylium to anti-adult T-cell leukemia cell activity.
The work is interesting since, after conduction a literature survey, this looks the first study involving metabolomics for functional comparison and molecular identification among Vaccinium virgatum plants. Also, it represents a continuation of their recent work focused on the quantitative analysis of water extracts of V. virgatum stems; ref. [16] of the present manuscript.
Overall, it is conducted with adequate means; however, English needs a moderate revision since some sentences need to be-rephased for better clarity e.g. lines 71-77.
Abstract. Delete: A fragment ion at m/z 149 was identified. Redundant!
1 Introduction. It is too maigre. Report more information on Vaccinium virgatum in botanical terms and also in the medical uses of the plant.
4.2. Extraction. Did the authors refer to a validated protocol?
- Conclusions. Need to be re-arranged. Avoid the term partial which may make confusion in the reader.
Author Response
Point 1: Overall, it is conducted with adequate means; however, English needs a moderate revision since some sentences need to be-rephased for better clarity e.g. lines 71-77.
Response 1: This manuscript has already twice received English proofreading by a native English speaker at Forte, Inc (Please see attachment file).
Point 2: Abstract. Delete: A fragment ion at m/z 149 was identified. Redundant!
Response 2: Thank you for your suggestion. According to your suggestion, we deleted "A fragment ion at m/z 149 was identified."
Point 3: 1 Introduction. It is too maigre. Report more information on Vaccinium virgatum in botanical terms and also in the medical uses of the plant.
Response 3: Another reviewer also suggested similar content. According to your suggestion, we made a significant revision of Introduction section (lines 38-44).
“Multiple species are involved in the commercial production of blueberries, with the vast majority composed of Vaccinium corymbosum L. (highbush blueberry) and its hybrids and V. angustifolium Aiton (lowbush blueberry), with lesser quantities of V. virgatum Aiton (rabbit-eye blueberry) [1,2]. V. virgatum is known to suitable for cultivation in warm regions [3]. Blueberry fruits are widely used in traditional medicine, pharmacy and the food industry [4-6]. The fruit of V. virgatum are edible and are used as sauces and syrups, and for bread, muffins, pancakes, and pies.”
Point 4: 4.2. Extraction. Did the authors refer to a validated protocol?
Response 4: Thank you for question. Initially, the extraction was performed as reported in the previous report [13], but the extract required for the experiment was not sufficiently obtained. Therefore, the temperature was raised by 10 degrees to extract. The ratio of plant amount to solvent amount is the same as in the previous report [13]. We added the below sentence. “The extraction method was slight modification of our previous report [13].”
Point 5: Conclusions. Need to be re-arranged. Avoid the term partial which may make confusion in the reader.
Response 5: Thank you for your suggestion. Another reviewer also suggested the same contents. As described below, we revised the conclusion to be a single sentence (lines 286-288).
“The results of this DI-EI-MS metabolomics analysis suggest that among PACs contained in V. virgatum stems and leaves, the fragment ion at m/z 149 contributes significantly to anti-ATL activity.”
Thank you very much for your kind suggestions.

Round 2
Reviewer 2 Report
The manuscript has been revised in accordance with the Reviewers' recommendations and is now suitable for publication in Plants.